# Prevalence and determinants of under-nutrition among children on ART in Ethiopia: A systematic review and meta-analysis

**Belete Gelaw Walle**[1]\*, **Nigusie Selomon**[2], **Chalie Marew Tiruneh**[2], **Bogale Chekole**[3], **Amare Kassaw**[2], **Moges Wubneh**[4], **Yibeltal Assefa**[5], **Kelemu Abebe**[6]

1 Department of Pediatric and Child Health Nursing, School of Nursing, College of Health Science and Medicine, Wolaita Sodo University, Wolaita Sodo, Ethiopia, 2 Department of Pediatric and Child Health Nursing, College of Medicine and Health Sciences, Debre Tabor University, Debre Tabor, Ethiopia, 3 Department of Pediatric and Child Health Nursing, College of Medicine and Health Sciences, Wolkite University, Wolkite, Ethiopia, 4 Department of Adult health Nursing, College of Medicine and Health Sciences, Debre Tabor University, Debre Tabor, Ethiopia, 5 School of Public Health, College of Health Science and Medicine, Wolaita Sodo University, Wolaita Sodo, Ethiopia, 6 School of Midwifery, College of Health Science and Medicine, Wolaita Sodo University, Wolaita Sodo, Ethiopia

\* beletegz12@gmail.com

**Data Availability Statement:** All relevant data are available within the paper and its Supporting information files.

## Abstract

### Background

Children living with HIV/AIDS are particularly vulnerable to under-nutrition. Under-nutrition associated with HIV/AIDS infection increases the rate of morbidity and mortality in children. To reaffirm a future objective, there needs to be evidence regarding the current national burden of under-nutrition and related factors among children infected with HIV. Hence, the objective of this systematic review and meta-analysis was to estimate the pooled prevalence of under-nutrition, and the pooled effect sizes of associated factors among HIV-infected children in Ethiopia.

### Methods

We searched Ethiopian universities' online libraries, Google, Google Scholar, PubMed, CINAHL, Cochrane Library, and Scopus to find the primary studies for this review. Publication bias was checked through Egger's regression test. Heterogeneity among the included studies was assessed using the $I^2$ test. The data were extracted using Microsoft Excel and exported to STATA Version 14 statistical software. A random effect meta-analysis model was performed to estimate the pooled prevalence of Under-nutrition.

### Results

After reviewing 1449 primary studies, 16 articles met the inclusion criteria and were included in the final meta-analysis. The estimated pooled prevalence of stunting, underweight, and wasting among children living with HIV/AIDS was 32.98% (95% CI: 22.47, 43.50), 29.76% (95% CI: 21.87, 37.66), and 21.16% (95% CI: 14.96, 27.35) respectively.

**Funding:** the authors received no specific funding for this work.

**Competing interests:** The authors have declared that no competing interests exist.

**Abbreviations:** AIDS, Acquired Immune Deficiency Syndrome; AOR, Adjusted Odd Ratio; ART, Antiretroviral Therapy; CI, Confidence Interval; COR, Crud Odd Ratio; HIV, Human immunodeficiency Virus; WHO, World Health Organization..

## Conclusions

This study showed that under-nutrition among HIV-infected children in Ethiopia was significantly high. Under-nutrition is more common among HIV-infected children with opportunistic infections, child feeding problems, do not adhere to dietary recommendations, and have diarrhea. The national policies and strategies for ART service- provider centers should maximize their emphasis on reducing under-nutrition among HIV-infected children. Based on this finding, we recommend HIV intervention programs to address nutritional assessment and interventions for HIV-infected children.

## Protocol registration

The protocol has been registered in the PROSPERO database with a registration number of CRD-394170.

## Introduction

Under-nutrition is the result of recurrent infectious diseases and insufficient food consumption. It includes stunting (being too short for one's age), wasting (being thin), being underweight for one's age, and micronutrient malnutrition (lacking of vitamins and minerals) [1]. Human immunodeficiency virus (HIV) and under-nutrition are highly interlinked diseases that lead to severe immune suppression [2]. Under-nutrition and HIV infection are still major public health concerns worldwide, particularly in sub-Saharan Africa. Both have the potential to weaken the immune system and make people more prone to infection, morbidity, and mortality [3, 4]. At the end of 2019, 38.0 million people were living with HIV worldwide, of which nearly 1.8 million were children (age 0-14years) [5]. Poverty, HIV infection, and food insecurity were found to be the leading causes of under-nutrition. Stunting, underweight, and wasting were more prevalent among HIV- positive children than HIV- negative children [6–8].

Globally, there were approximately 50 million wasted and 156 million stunted children [9]. Under-nutrition is a well-known indicator of infants and young children's nutritional status in low and middle-income countries [10]. Under-nutrition and HIV infection increase the risk of child growth failure, morbidity, and mortality [3, 11]. The magnitude of stunting, wasting, and underweight are higher among HIV- infected children as compared with non-infected children [6, 12, 13]. In developing countries, HIV infection has exacerbated the effects of under-nutrition on children [14].

The rate of HIV-related mortality has been reduced since the launch of antiretroviral therapy (ART) [15]. However, ART can cause metabolic disorders and have a negative impact on children's nutritional status, especially during the first months of treatment [16]. The problems will interact (double burden) among HIV- infected children with under-nutrition. As a result, providing ART service without nutritional management, or nutritional management without ART service, may result in poor treatment outcomes [17].

Globally, under-nutrition is responsible for 35% of all child deaths, 11% of the disease burden and the risk of death is tripled in HIV- infected children [18, 19]. Under-nutrition has devastating health and socio-economic consequences at the household, individual, community, and national level [18, 20]. Developing countries or economically disadvantaged regions carried the highest burden of under-five mortality, accounting for more than one-third of total under-five annual mortality due to under-nutrition [11].

Despite Ethiopia's initiatives to combat food insecurity and poverty, many areas remain vulnerable to food insecurity and malnutrition [21]. Even after interventions by stakeholders, child mortality from malnutrition remains a major public health concern, especially in developing countries such as Ethiopia. In Ethiopia, different studies [3, 8, 12, 22–31] showed the prevalence of under nutrition with great inconsistencies across different geographical regions at different periods of time. Identifying modifiable risk factors for under-nutrition among HIV-infected children is a critical step in identifying potential interventions.

Furthermore, reliable, updated, and summarized nationwide pooled data on under-nutrition among HIV-infected children is necessary to improve government policies, strategies, and interventions. Therefore, this systematic review and meta-analysis aimed to estimate the pooled national prevalence of under-nutrition and effect size of factors associated with under-nutrition among HIV- infected children using currently available articles in Ethiopia. The findings from this review will highlight the prevalence and factors of under-nutrition with implications for improving health workers' interventions, resource allocation, and for accelerating the reduction of under-nutrition among HIV infected children in Ethiopia.

## Materials and methods

### Reporting

The findings of this meta-analysis and systematic review were organized and reported through the Preferred Reporting Items for Systematic Reviews and Meta-Analyses (PRISMA) guideline [32].

### Study design, settings and search strategies

A systematic review and meta-analysis was done to estimate the pooled prevalence under-nutrition and pooled effect size of factors associated with under-nutrition among HIV-infected children in Ethiopia. Ethiopia is originated in the Horn of Africa. Ethiopia is bounded by Sudan and South Sudan to the west, Kenya to the south, Eritrea to the north, and Djibouti and Somalia to the east. The published and unpublished/grey literature describing the prevalence and associated factors of under-nutrition (underweight, stunting, and wasting) among HIV-infected children were reviewed. Both manual and electronic searches were used to find potentially relevant articles. The Population, Exposure, Comparison, and Outcomes (PECO) search formula was used in this review to search pertinent articles. All eligible HIV-infected children were the population of interest for this study. The predictors or determinants of under-nutrition included in this study were exposures. Comparisons were the reported reference group for each predictor or determinant in each respective study. The outcome of interest was under-nutrition among HIV-infected children. For each of the selected components of PECO, an electronic database search was done through the medical subject heading [MeSH] words and the keyword search.

Relevant articles were searched from Google for gray literature., Google Scholar, PubMed, CINAHL, Cochrane Library, and Scopus. MeSH (Medical Subject Headings), Boolean operators and all fields within records were used to search in the advanced PubMed search engine. The key words used for the advanced PubMed search strategy were ((((((((((((Prevalence [tw] OR magnitude [tw]) OR ("Prevalence"[MeSH Terms] OR "magnitude"[MeSH Terms] OR "incidence"[MeSH Terms])) AND (undernutrition [tw] OR nutritional deficiency [tw] OR undernutrition[tw] OR malnourishment[tw] OR stunting[tw] OR wasting[tw] OR underweight[tw])) OR ("malnutrition"[MeSH Terms] OR "growth disorders"[MeSH Terms] OR "cachexia"[MeSH Terms] OR "thinness"[MeSH Terms])) AND (factors[tw] OR determinants [tw] OR risk factors[tw])) OR (" factors"[MeSH Terms] OR "risk factors"[MeSH Terms] OR "

determinants "[MeSH Terms])) AND (Children[tw] OR Infant[tw])) OR ("child"[MeSH Terms] OR "infant"[MeSH Terms])) AND (Antiretroviral therapy[tw] OR ART [tw])) OR (("anti-retroviral agents"[All Fields] OR "anti-retroviral agents"[MeSH Terms] OR "therapeutics"[MeSH Terms])) AND (Human Immunodeficiency Virus [tw] OR HIV [tw] OR AIDS)) OR ("HIV"[MeSH Terms] OR "acquired immunodeficiency syndrome"[MeSH Terms])) AND (Ethiopia [tw]). The search of the articles was done by YA and BGW. All primary articles reporting the prevalence of at least one underweight, wasting, or stunting and published in English up to March 31, 2023 were included.

## Measurement outcome variables

This review has two main objectives. The first objective of the study was to estimate the pooled prevalence of under-nutrition among HIV-infected children in Ethiopia. The second objective was to identify factors associated with under-nutrition (stunting, underweight and wasting) among HIV- infected children in Ethiopia. The prevalence of childhood under-nutrition was estimated by dividing the total number of study participants by the outcome of interest to the overall children participating in the study multiplied by 100. In the included primary studies, the prevalence of stunting, underweight, and wasting was measured using height for age- Z score (HAZ), weight for age -Z score (WAZ), and weight for height -Z score (WHZ), <-2sd respectively [33]. For the second objective, we determined the association between under-nutrition and associated factors in the form of the log odds ratio.

## Eligibility criteria

**Inclusion criteria and exclusion criteria.** Both published and unpublished observational studies reporting the prevalence and associated factors of under-nutrition among HIV-infected children in Ethiopia were considered. However, studies whose study participants were either adults or both adults and children were excluded. Besides, studies that reported neither the prevalence nor factors associated with under-nutrition (stunting, underweight and wasting) among HIV- infected children weren't eligible for this review. Conference reports and articles without full text access were also excluded. The exclusion of these studies was because of the inability to check the quality of the studies in the absence of full text. For further clarity, the topic is described using a PICO format as follows:

**Population (P)**: HIV-infected children
**Intervention (I)**: Children living with HIV
**Comparison (C)**: Children living without HIV
**Outcome (O)**: Under-nutrition.

## Data extraction process

The required data from the included primary articles were extracted by two reviewers (BGW and KA) using a standardized data extraction format, adapted from the Joanna Briggs Institute (JBI). The data extraction sheet was piloted using six randomly selected papers, and adjustments were made after the template was piloted. Two reviewers assessed the retrieved articles for inclusion using their title, abstract, and full text. Any disagreements during screening were undertaken through discussion and consensus (Delphi technique). For the first outcome variable (prevalence), the data extraction format included the primary author name, study area, publication year, region(s) of the country where the study was done, study design, sample size, response rate, and prevalence (stunting, underweight, wasting) with a 95% CI. For the second outcome variable (factors), data were extracted in the format of two by two tables, and then the

log odds ratio for each factor associated with stunting, underweight, or wasting was calculated based on the findings of the primary studies.

### Risk of bias assessment

Two authors (BGW and KA) independently assessed the included articles for risk of bias using Hoy's [33] bias assessment tool, which consists of ten criteria with four domains of bias to evaluate internal and external validity. The first four criteria evaluate the presence of selection bias, nonresponse bias, and external validity. The next six criteria evaluate the presence of measuring bias, analysis-related bias, and internal validity. Each article was classified as either low, moderate, or high risk of bias. Finally, the score of risk of bias was categorized based on the number of "yes" per study: low risk ($\geq 8$), moderate risk (6–7), and high risk of bias ($\leq 5$) (S1 Table).

### Quality assessment

Two investigators (BGW and KA) appraised the qualities of the included articles using the Newcastle-Ottawa Scale quality assessment tool for observational studies [34]. The tool has three sections, in which the first section focuses on the methodological quality of each original study (i.e., sample size, response rate, and sampling technique) and is graded out of five stars. The second section considers the comparability of the study cases or cohorts, with a probability of two stars to be gained. The third section concerns outcomes and statistical analysis of the original study, with the possibility of three star scores. Any inconsistencies between the two quality evaluators were resolved by repeating the procedures and then taking the average of the two assessment scores. Articles with a total score of $\geq 6$ from 10 and 9 for the cross-sectional and cohort studies were considered to be of high quality, respectively (**S2 Table**).

### Data processing and statistical analysis

The required data from each primary study was extracted using Microsoft Excel spreadsheet format before analysis. The data were exported to STATA Version 14 statistical software for analysis. The level of heterogeneity between primary studies was assessed through the $I^2$ test [35]. A random-effects meta-analysis model was employed to estimate the pooled prevalence. To identify the possible source of heterogeneity, sub-group analysis was carried out using the region(s) of the country, study design, and year of publication. Publication bias across studies was assessed using a funnel plot (visual inspection) and Egger's regression test at the 5% significant level [36, 37].

## Results

### Database search results

Initially, 1449 primary articles were retrieved reporting prevalence and factors associated with under-nutrition among HIV-infected children using the range of databases previously described. Of these initial articles, 664 were excluded due to duplications. From the remaining articles, 253 were excluded after review of their titles and abstracts, and 309 were excluded due to different reasons. Then, 223 full-text articles were assessed for eligibility based on the preset criteria, and 207 articles were further excluded due to different reasons. Finally, 16 articles met the eligibility criteria and were included in the final meta-analysis to determine the prevalence and associated factors of under-nutrition (**Fig 1**).

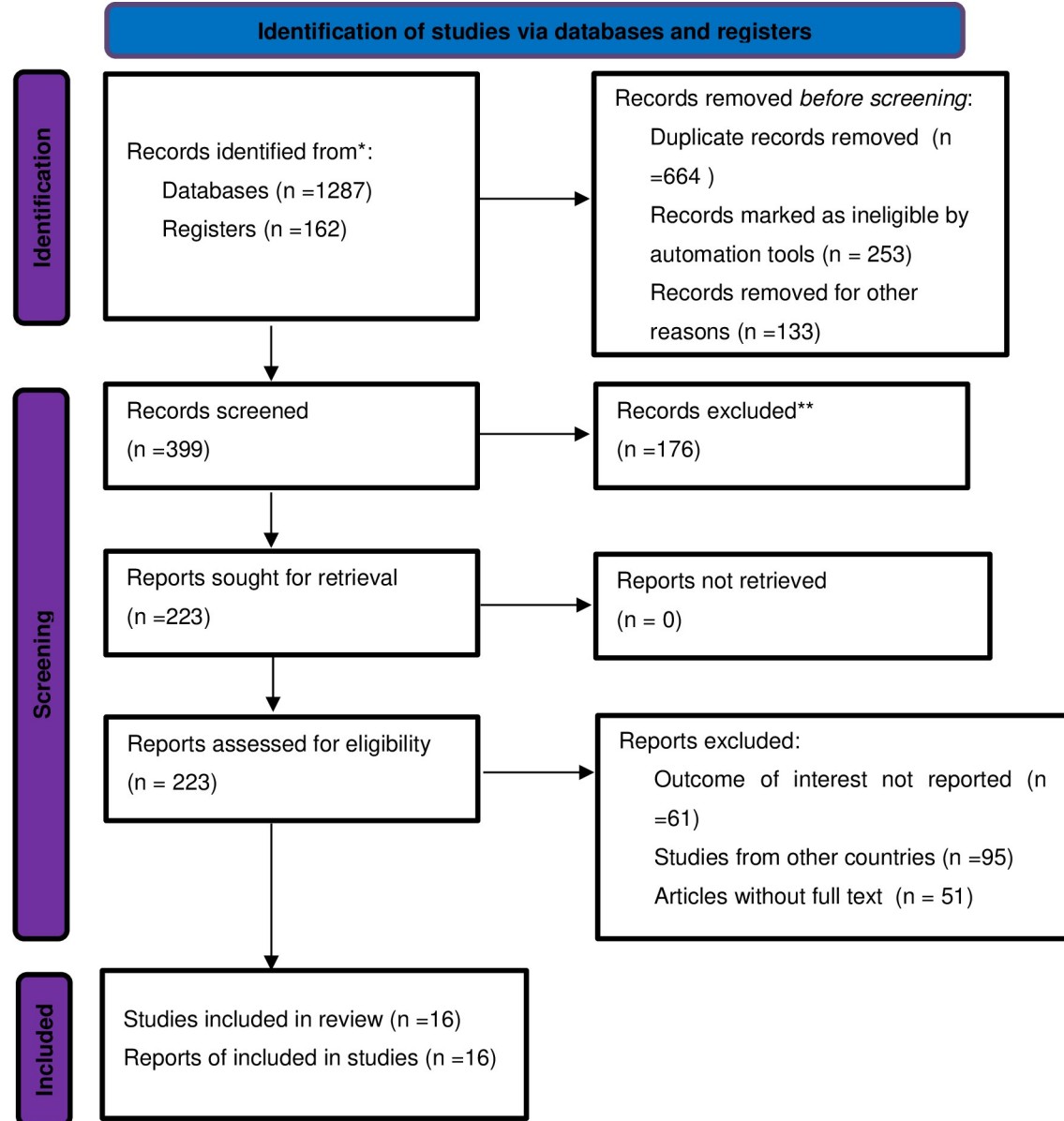

**Fig 1. PRISMA 2020 flow chart of primary study selection for systematic review and meta-analysis of under nutrition among HIV infected children in Ethiopia, from 2012–2022.**

### General characteristics of included studies

As described in Table 1, among the included studies, 7 were done in the Amhara region [4, 23–25, 27, 31, 38], whereas 2 in Oromia [22, 30], 1 in Addis Ababa [39], 2 in Eastern Ethiopia [3, 12], 1 in Harari [26], 2 from Sidama [28, 29], and 1 in the Southern Nations, Nationalities, and Peoples (SNNP) [8]. Regarding study design and publication year, most studies (87.5%) were cross-sectional, and published from 2012 to 2022, respectively. In this meta-analysis study, 5,790 study participants were involved to determine the pooled prevalence of under-nutrition among HIV-infected children. The sample size of the studies fluctuated between 108 [26] and 732 [40]. Concerning the methodological quality of the included studies, 13 studies

**Table 1. General characteristics of studies included in systematic review and meta-analysis of under-nutrition and associated factors among HIV-infected children in Ethiopia, from 2012–2022.**

| Author/s (reference) | Study Design | Publication Year | Study area, Region | Sample size | Response rate (%) | Stunting (%) | Wasting (%) | Under-weight (%) | Quality |
|---|---|---|---|---|---|---|---|---|---|
| Haileselassie et al | Cross-sectional | 2019 | Eastern Ethiopia | 390 | 96.60 | 24.70 | 28.20 | | Low risk |
| Tiruneh et al | Cross-sectional | 2021 | SNNP | 406 | 94.30 | 5.50 | 36.30 | | Low risk |
| Gezahegn et al | Cross-sectional | 2020 | Eastern Ethiopia | 414 | 97 | 30.90 | | | Low risk |
| Jeylan et al | Cross-sectional | 2018 | Oromia | 412 | 100 | 13.40 | | 21.80 | Low risk |
| Megabiaw et al | Cross-sectional | 2012 | Amhara | 301 | 100 | 65.00 | 5.8 | 41.70 | Low risk |
| Kedir et al | Cohort | 2014 | Oromia | 560 | 72.70 | | | 51.60 | Low risk |
| Abdulkadir | Cross-sectional | 2014 | Amhara | 142 | 100 | 46.50 | 31.70 | 40.80 | Low risk |
| Tiruneh et al | Cross-sectional | 2022 | Amhara | 406 | 93.30 | | | 28.00 | Low risk |
| Teklemariam et al | Cross-sectional | 2015 | Harari | 108 | 100 | 15.90 | 9.80 | 8.90 | Low risk |
| Mengist et al | Cross-sectional | 2022 | Amhara | 270 | 100 | 10.70 | 27.00 | 15.90 | Low risk |
| Kusum Lata | Cross-sectional | 2020 | Sidama | 455 | 92.30 | 60.20 | 21.40 | 41.20 | Low risk |
| Shiferaw and Gebremedhin | Cross-sectional | 2020 | Sidama | 260 | 100 | 33.10 | 20.00 | | Low risk |
| Sewale et al | Cross-sectional | 2021 | Amhara | 372 | 100 | 45.20 | | | Low risk |
| Tekleab et al | Cohort | 2016 | Addis Abeba | 202 | 100 | 71.30 | 16.30 | 39.50 | Low risk |
| Kebede et al | Cohort | 2022 | Amhara | 732 | 98.50 | 21.36 | 16.50 | 19.97 | Low risk |
| Dessalegn et al | Cross-sectional | 2021 | Amhara | 360 | 88.90 | 19.40 | | 19.20 | Low risk |

[3, 8, 12, 22–29, 31, 38] were evaluated based on the JBI check list for cross-sectional studies, and the remaining 3 studies [30, 39, 40] based on the JBI check list for cohort studies. Regarding to response rate, almost all articles had good response rate (>85%), which may, in part be attributable to the use of interviewer-administered questionnaires to collect the data (**Table 1**).

## Meta-analysis of under-nutrition

**Prevalence of under-nutrition.** A random effect meta-analysis model was computed to estimate the pooled prevalence of under-nutrition among HIV-infected children in Ethiopia. The pooled prevalence of stunting among HIV-infected children in Ethiopia was found to be 32.98% (95% CI: 22.47, 43.50, $I^2$ = 98.90%, p<0.01) (**Fig 2**).

Similarly, this study revealed that the pooled prevalence of wasting among HIV-infected children in Ethiopia was 21.16% (95% CI: 14.96, 27.35, $I^2$ = 95.50%, p<0.01) (**Fig 3**).

Besides, the overall pooled prevalence of underweight among HIV-infected children was 29.76% (95% CI: 21.87, 37.66, $I^2$ = 97%, p<0.01) (**Fig 4**).

There was significant variation across the included original studies to estimate the prevalence of stunting, underweight and wasting. In sensitivity analysis, the prevalence of stunting,

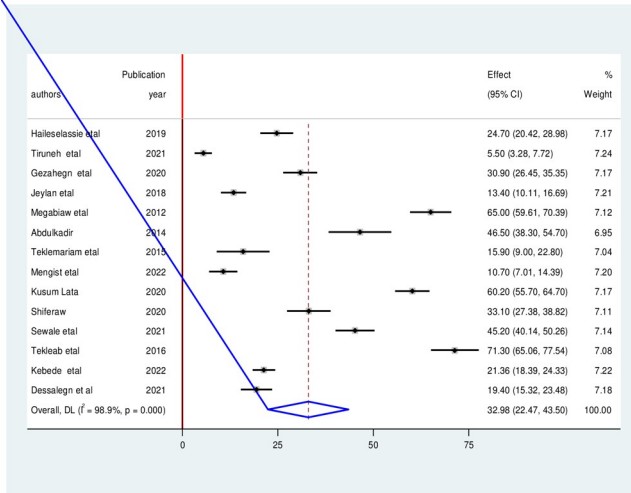

**Fig 2. Forest plot of the pooled prevalence of stunting among HIV- infected children in Ethiopia, from 2012–2022.**

wasting, and underweight varied from 31.21 (95% CI: 20.68, 41.73) [39] to 36.70 (95% CI: 25.72, 47.67) [8] (**S3 Table**), 19.18 (95% CI: 12.84, 25.52) [8] to 23.12 (95% CI: 17.11, 29.13) [23] (**S4 Table**), and from 25.86 (95% CI: 19.57, 32.15) [28] to 29.49 (95% CI: 22.86, 36.12) [26] (**S5 Table**), respectively after the deletion of a single study. Accordingly, there is no study away from the lower and upper limits of the confidence interval, which confirms that there is no influential study.

   **Risk of publication bias in studies.**   Publication bias was assessed using the Egger test, which showed no possibility of statistically significant publication bias for stunting (p-value 0.09), wasting (p-value 0.08), and underweight (p-value 0.43), respectively. We also performed publication bias assessment using funnel plots, which have slight asymmetrical distributions for stunting (**Fig 5**), wasting (**Fig 6**), and underweight (**Fig 7**), indicating publication bias.

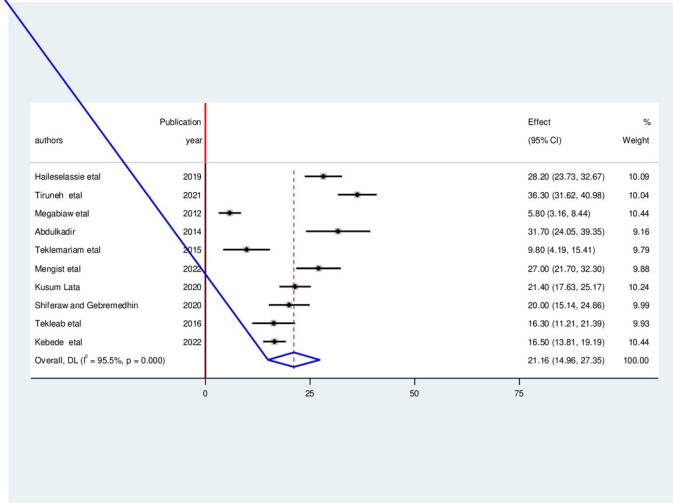

**Fig 3. Forest plot of the pooled prevalence of wasting among HIV- infected children in Ethiopia, from 2012–2022.**

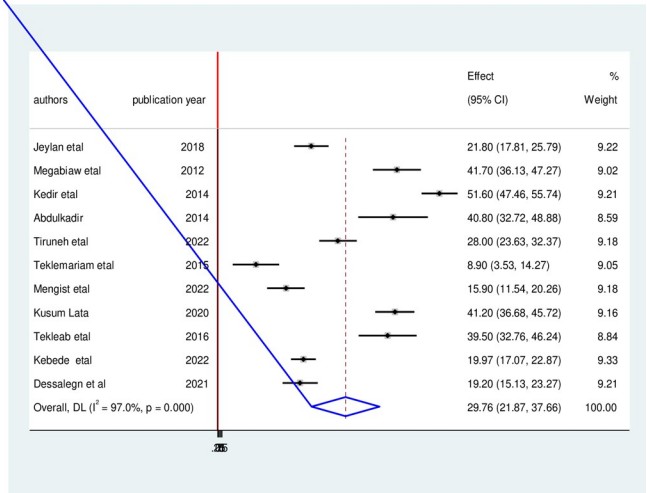

**Fig 4. Forest plot of the pooled prevalence of under-weight among HIV- infected children in Ethiopia, from 2012–2022.**

**Subgroup analysis.** In this review, we performed subgroup analysis based on the region of the country where studies were conducted, study design, and publication year to identify the source of heterogeneity. Accordingly, the highest prevalence of stunting was observed in Sidama with a prevalence of 46.71% (95% CI: 20.15, 73.27) and the lowest in eastern Ethiopia at 27.77% (95% CI: 21.69, 33.84) (**Table 2**). The highest prevalence of underweight was observed in the Oromia Region, with a prevalence of 36.69% (95% CI: 7.49, 65.90), and the lowest in Amhara, 27.18% (95% CI: 19.72, 34.63) (**Table 2**). Additionally, the highest prevalence of wasting was in Sidama, 20.87% (95% CI: 17.90, 23.85), and the lowest was in Amhara Region, 19.81% (95% CI: 9.71, 29.91) (**Table 2**). Regarding prevalence by study design, stunting among HIV-infected children was 30.79% (95% CI: 19.533, 42.06), 46.25% (95% CI: -2.69,

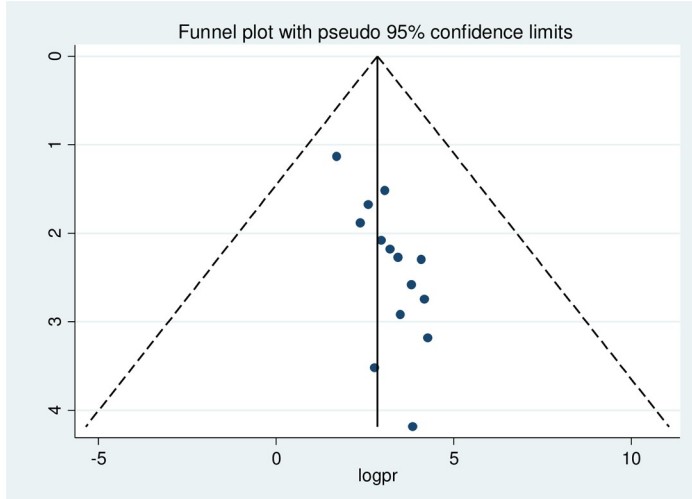

**Fig 5. Funnel plot with 95% confidence limits of the pooled prevalence of stunting among HIV- infected children in Ethiopia, from 2012–2022.**

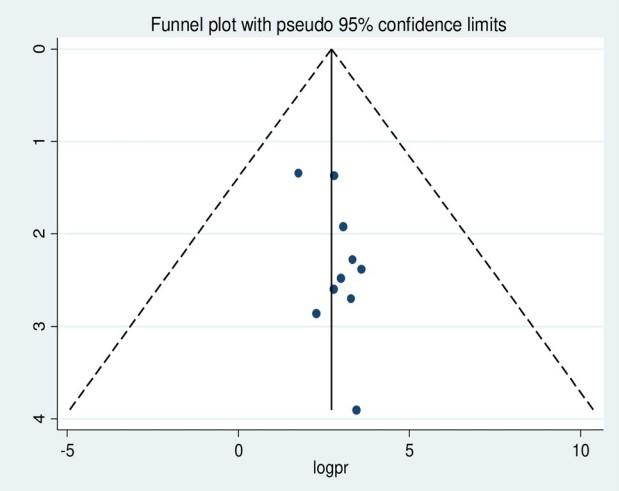

**Fig 6. Funnel plot with 95% confidence limits of the pooled prevalence of wasting among HIV- infected children in Ethiopia, from 2012–2022.**

95.19), from cross-sectional and cohort studies respectively (**Table 2**). On the other hand, the overall prevalence of wasting was found to be 22.42% (95% CI: 14.10, 30.73) in cross-sectional studies (**Table 2**) and 16.46% (95% CI: 14.08, 18.83) in cohort studies (**Table 2**). The pooled prevalence of underweight among HIV-infected children was 27.03% (95% CI: 18.92, 35.15) in cross-sectional studies (**Table 2**) and 36.97% (95% CI: 15.02, 58.93) in cohort studies (**Table 2**). The overall prevalence of stunting among HIV-infected children was 42.51% (95% CI: 12.24, 72.78) in studies published from 2012 to 2015 and 30.41% (95% CI: 19.45, 41.37) in studies published from 2016 to 2022 (**Table 2**). The prevalence of wasting among HIV-infected children was 15.33% (95% CI: 2.31, 28.35) in studies published from 2012 to 2015, whereas it was 23.60% (95% CI: 18.22, 28.97) in studies published from 2016 to 2022 (**Table 2**). The

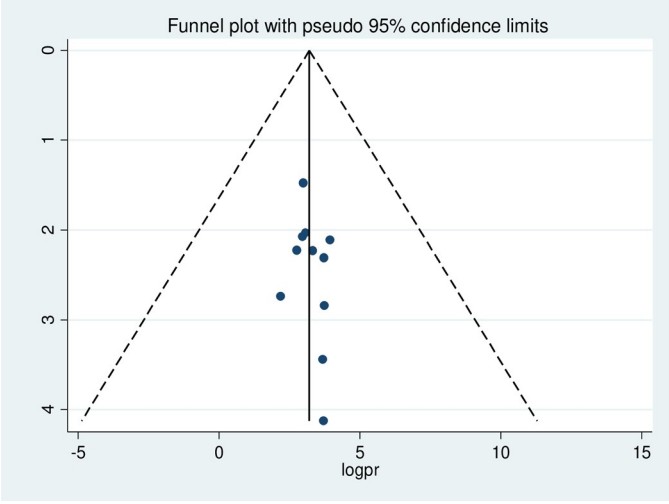

**Fig 7. Funnel plot with 95% confidence limits of the pooled prevalence of under-weight among HIV- infected children in Ethiopia, from 2012–2022.**

**Table 2. Summary of subgroup analysis for the pooled prevalence of under-nutrition among HIV-infected children in in Ethiopia, from 2012–2022 (n = 16).**

| Variables | Subgroup | Stunting,%(95%CI, $I^2$, P-value) | Wasting,%(95% CI, $I^2$, P-value) | Underweight,%(95%CI, $I^2$, P-value) |
|---|---|---|---|---|
| Region | Amhara | 34.55(19.25, 49.84,98.6, < 0.01) | 19.81(9.71,29.91,96.4, < 0.01) | 27.18(19.72,34.63,94.10, < 0.01) |
| | Eastern Ethiopia | 27.77(21.69, 33.84,74.2, 0.049) | 28.20(23.73,32.67,100,-) | 8.90(3.53,14.277,-) |
| | SNNP | 5.50 (3.28, 7.72,-) | 36.30(31.62,40.98,-) | 41.20(36.68,45.72,-) |
| | Oromia | 13.40 (10.11, 16.69,-) | - | 36.69(7.49,65.90,99,< 0.01) |
| | Harari | 15.90 (9.00, 22.80,-) | 9.80(4.19,15.41,-) | - |
| | Sidama | 46.71 (20.15, 73.27,-) | 20.87(17.90,23.85,-,0.656) | - |
| | Addis Abeba | 71.30 (65.06, 77.54,-) | 16.30(11.21,21,39,-) | 39.50(32.76,46.24,-) |
| Study design | Cross-sectional | 30.79(19.53,42.06,98.8, < 0.01) | 22.42(14.10,30.73,96.4,< 0.01) | 27.03(18.92,35.15,95.70, < 0.01) |
| | Cohort | 46.25(2.69,95.19, 99.5, < 0.01) | 16.4(14.08,18.83,-,0.946) | 36.97(15.02,58.93,98.70, < 0.01) |
| Publication year | 2012–2015 | 42.51(12.24,72.78,98.3, < 0.01) | 15.33(2.31,28.35,94.90, < 0.01) | 35.75(15.82,55.68,98.10, < 0.01) |
| | 2016–2022 | 30.41(19.45,41.37,98.9, < 0.01) | 23.60(18.22,28.97,91.2, < 0.01) | 26.30(19.65,32.96,94.40, < 0.01) |

pooled prevalence of underweight among HIV-infected children was found to be 35.75% (95% CI: 15.82, 55.68) in studies published from 2012 to 2015 (**Table 2**) and 26.30% (95% CI: 19.65, 32.96) in studies published from 2016 to 2022 (**Table 2**).

**Associated factors of under-nutrition.** In this meta-analysis, the associated factors were categorized into three thematic areas. These were: 1. factors associated with stunting; 2. factors associated with wasting; and 3. factors associated with underweight.

**Factors associated with stunting.** In this review, we examined the association between variables and childhood stunting by using 14 primary studies [3, 12, 22–29, 31, 38–40]. For this study, at least three primary studies with factors significantly associated with stunting were incorporated. As shown in Figs 4 and 5, the occurrence of stunting was significantly associated with opportunistic infections (OI) and child feeding problems. The results of these seven primary studies indicated that stunting was significantly associated with OI [3, 8, 12, 22, 24, 29, 31]. The pooled odds ratio indicated that the likelihood of stunting occurrence was 3.15 times higher among children who were infected with any of the OIs as compared to their counterparts (OR: 3.15, 95%CI: 1.85, 5.33, I2 = 99.3%, P<0.01) (**Fig 8**).

In addition, to determine the association between children with feeding problems and stunting, four primary studies were included in the analysis [8, 22, 29, 31]. The pooled odds ratio revealed that children with feeding problems were 2.55 times more likely to develop stunting than their counterparts (OR: 2.55, 95%CI: 1.87, 3.48, I2 = 96.4%, P<0.01) (**Fig 9**).

**Factors associated with wasting.** In the meta-analysis of three primary studies [3, 28, 29], the odds of wasting among HIV-infected children who had diarrhea were 2.6 times higher than children who had no diarrhea (OR: 2.60, 95%CI: 1.55, 4.37, I2 = 99.3%, P<0.01) (**Fig 10**).

Besides, three primary studies were included in this meta-analysis to determine the association between child feeding problems and the occurrence of wasting [3, 22, 23]. Accordingly, children with feeding problems were significantly associated with wasting among HIV-infected children in Ethiopia (OR: 2.25, 95% CI: 2.07, 2.44, I2 = 53.70%, P = 0.115) (**Fig 11**).

**Factors associated with under-weight.** In this meta-analysis, we examined the association between dietary counseling and childhood underweight among HIV-infected children using three original studies [23, 25, 28]. The results from these three studies revealed that the occurrence of childhood underweight was significantly associated with dietary counseling status. Consequently, the likelihood of childhood underweight occurrence was 3.93 times higher among HIV-infected children who did not adhere to dietary counseling as compared to their counterparts (OR: 3.93, 95%CI: 1.97, 7.85, I2 = 95.20%, P<0.01) (**Fig 12**).

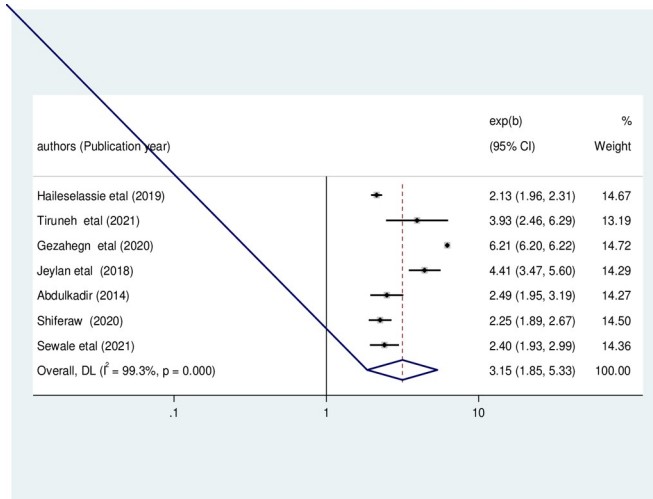

**Fig 8. The pooled odds ratio of the association between opportunistic infection and stunting among HIV-infected children in Ethiopia, from 2012–2022.**

Similarly, three included primary studies [22, 23, 28] reported that the family monthly income level was not significantly associated with childhood underweight among HIV-infected children in Ethiopia (OR: 0.80, 95%CI: 0.42, 1.52, I2 = 98.20%, P<0.01) (**Fig 13**).

## Discussion

Under-nutrition is one of the major causes of morbidity and mortality among HIV-infected children in Ethiopia. Though, the proportion of under-nutrition, particularly among HIV-infected children, has not been fully explored. This meta-analysis was conducted to estimate the national pooled prevalence of under-nutrition (stunting, wasting, and underweight) and associated factors among HIV-infected children in Ethiopia. Hence, estimating the pooled

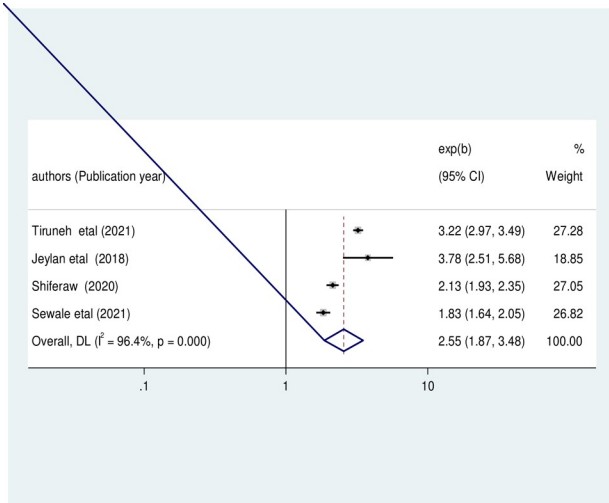

**Fig 9. The pooled odds ratio of the association between child feeding problem and stunting among HIV- infected children in Ethiopia, from 2012–2022.**

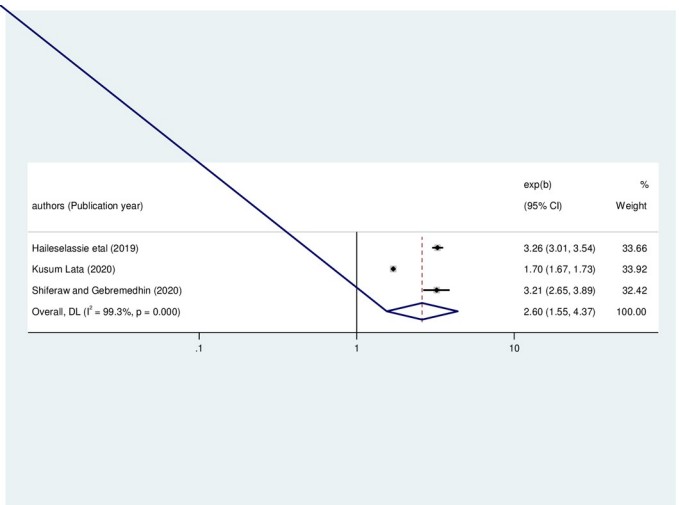

**Fig 10. The pooled odds ratio of the association between childhood diarrhea and wasting among HIV- infected children in Ethiopia, from 2012–2022.**

prevalence of under-nutrition and its contributing factors in Ethiopia could have value for policymakers and clinicians to take corrective action. The nationally pooled estimate of stunting, wasting, and underweight was found to be 32.98% (95% CI: 22.47, 43.50), 21.16% (95% CI: 14.96, 27.35), and 29.76% (95% CI: 21.87, 37.66) respectively. Besides, opportunistic infection, child feeding problems, diarrhea, and dietary counseling adherence were statistically significant factors having positive odds of association with under-nutrition among HIV-infected children in Ethiopia.

The result of this meta-analysis revealed that the pooled prevalence of stunting among HIV-infected children in Ethiopia was 32.98% (95% CI: 22.47, 43.50). The result of this meta-analysis was in line with studeis done on HIV-infected children in Central and West Africa

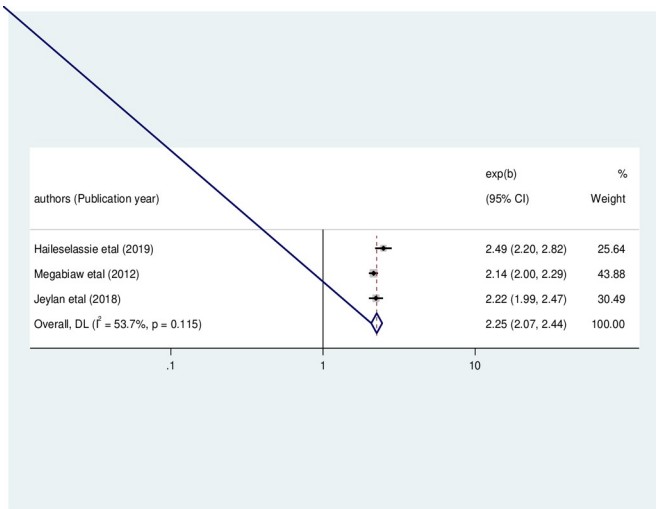

**Fig 11. The pooled odds ratio of the association between child feeding problem and wasting among HIV- infected children in Ethiopia, from 2012–2022.**

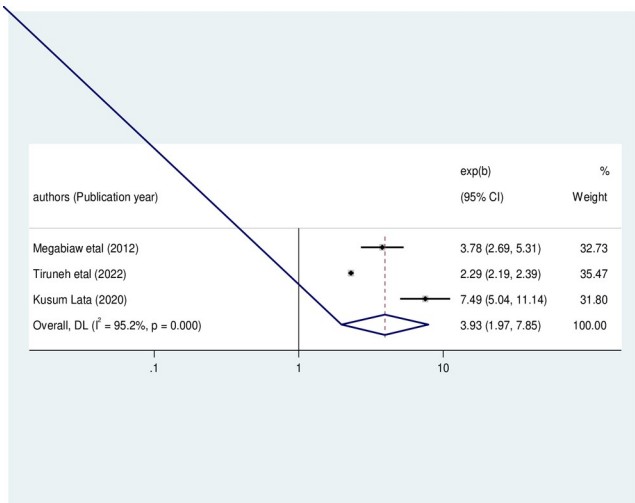

**Fig 12. The pooled odds ratio of the association between dietary counseling adherence and under-weight among HIV- infected children in Ethiopia, from 2012–2022.**

(33%) [41], WHO report (32.5%) [42], and study conducted in developing region (41%) [43]. However, this finding was much lower than a meta-analysis conducted in southern Africa, which found the prevalence of stunting at 61.1% [44] and in south India at 58% [45]. Our finding was also lower than the meta-analysis conducted among HIV-infected children in African countries, which estimated the prevalence of stunting among HIV-infected children to be 46.7% [46] and 49.68% [47]. The possible explanation for the above variation could be due to the differences in the study subjects' socio-demographic characteristics and diagnostic approaches to stunting in the Ethiopian context and other countries.

In this meta-analysis, the pooled proportion of wasting among HIV–infected children in Ethiopia was 21.16% (95% CI: 14.96, 27.35). This overall proportion is consistent with a

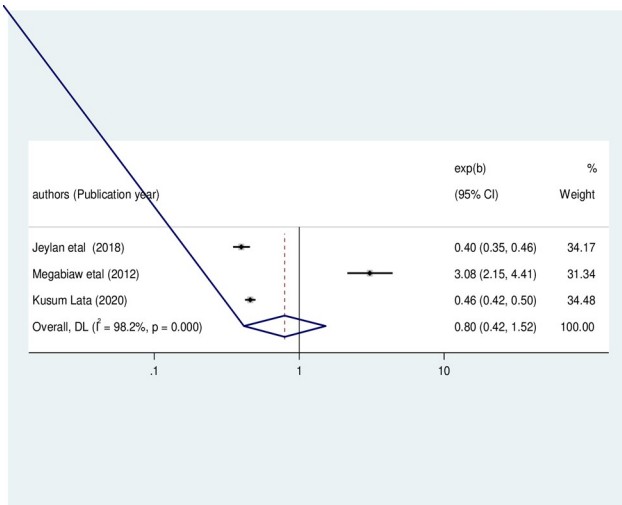

**Fig 13. The pooled odds ratio of the association between family monthly income and under-weight among HIV-infected children in Ethiopia, from 2012–2022.**

systematic review conducted in sub-Saharan Africa, which showed the prevalence of wasting ranged between 18.67% and 27.42% [46], as well as a study done in west and central African countries 16% [41]. The finding is also in agreement with a systematic review conducted in east Africa, which suggests a prevalence of 24.65% [47] and a study conducted in southern Africa, which found a prevalence of 21.33% [44]. This finding is higher than studies conducted in the industrialized world, which revealed a 14.5% prevalence of wasting among children [43], and the WHO reported a 6.4% prevalence [42]. The possible explanation for the above variation could be explained by the differences in socio-demographics, socio-economics, and socio-cultural practices, which have a great impact on child health status, responses to disease, and treatment outcomes. Another possible explanation might be related to the difference in concerns and policies of governmental and non-governmental stakeholders towards the impact of HIV on children's nutritional status.

According to this meta-analysis, the pooled prevalence of underweight among HIV-infected children in Ethiopia was 29.76% (95% CI: 21.87, 37.66). This finding is in line with a previous meta-analysis conducted in sub-Saharan African countries (35.9%) [46]. This finding is nearly twice as high as a study done in Nigeria [48]. Conversely, our finding is lower than a meta-analysis conducted in east Africa (41%) [47], a study done in southern Africa (47.3%) [44], and a study done in south India (65%) [45]. The reason for the discrepancy may be due to the differences in geographical areas, sample size, and cultures of study participants.

In this meta-analysis, we also explored factors associated with under-nutrition among Ethiopian HIV-infected children. The results indicated that opportunistic infections (OI) and child feeding problems were significantly associated with stunting. Diarrhea and feeding problems were found to be factors significantly associated with wasting. Dietary counseling adherence was significantly associated with childhood underweight among HIV-infected children in Ethiopia.

This meta-analysis revealed that children with opportunistic infections (OI) were 3.15 times more likely to be stunted than their counterparts. HIV/AIDS can have a negative impact on the immune system, resulting in severe immune deficiency, which increases susceptibility to infection, lowers food intake, and causes malnutrition. Children with feeding problems were 2.55 times more likely to develop stunting than their counterparts. The reason might be due to micronutrient deficiency and starvation from feeding problems associated with HIV disease, which may lead to stunting. Moreover, the odds of wasting among HIV-infected children who had diarrhea were 2.6 times higher than those who did not. This might be due to decreased food consumption (poor appetite, inability to eat and swallow), increased energy requirements, and decreased nutrient absorption in the child. The odds of wasting among HIV-infected children who had feeding problems were also 2.25 times higher than those who did not have a feeding problem. The reason could be that feeding problems often lead to nutritional deficiencies, which in turn can hasten stunting.

Finally, the likelihood of childhood underweight occurrence was 3.93 times higher among HIV-infected children who did not adhere to dietary counseling compared to their counterparts. This might be due to the fact that unhealthy eating habits can negatively affect the overall quality of life, immunological function, and the effectiveness of antiretroviral therapy. So that dietary counseling non-adherence may expose children to opportunistic infections and affect the weight of HIV-infected children. As mothers are the most frequent primary caregivers for their children, it is important to assess their child feeding practices and child antiretroviral therapy adherence. HIV-infected children require more attention throughout their follow-up as they are more vulnerable to under-nutrition (stunting, wasting, and underweight).

## Limitations of the study

This national review included only articles or reports studied in the English language, which may limit some papers from being included. Besides, this study represented only studies conducted in six regions and one administrative town of the country, which might be under-representation due to the limited number of studies included. The majority (87.5%) of the studies included were cross-sectional; subsequently, the outcome variables may be affected by other confounding variables. Moreover, the majority of the studies included in this meta-analysis had a small sample size, which could affect the estimated prevalence of under-nutrition.

## Conclusion

In this meta-analysis, under-nutrition among HIV-infected children in Ethiopia was found to be significantly high. Opportunistic infection (OI) and child feeding problems were significantly associated with stunting. Dietary counseling adherence was significantly associated with underweight. Likewise, diarrhea and feeding problems were significantly associated with the occurrence of wasting among HIV-infected children in Ethiopia. Thus, based on our results, we recommend that particular emphasis shall be given to prevent under-nutrition and improve treatment adherence, which in turn helps to hasten treatment outcomes.

## Supporting information

**S1 Checklist. PRISMA 2020 checklist.**
(DOCX)

**S1 Table. Risk of bias assessment for the included studies using the Hoy 2012 tool.**
(DOCX)

**S2 Table. Shows the quality score of each study using the Newcastle-Ottawa Scale (NOS) quality assessment tool adapted for cross-sectional and cohort studies.**
(DOCX)

**S3 Table. Sensitivity analysis for the pooled prevalence of stunting in Ethiopia, from 2012–2022.**
(DOCX)

**S4 Table. Sensitivity analysis for the pooled prevalence of wasting in Ethiopia, from 2012–2022.**
(DOCX)

**S5 Table. Sensitivity analysis for the pooled prevalence of Under-weight in Ethiopia, from 2012–2022.**
(DOCX)

## Acknowledgments

We would like to express our special gratitude to the authors of primary studies that were incorporated into this systematic review and meta-analysis.

## Author Contributions

**Conceptualization:** Belete Gelaw Walle.

**Data curation:** Belete Gelaw Walle, Nigusie Selomon, Chalie Marew Tiruneh, Bogale Chekole, Amare Kassaw, Moges Wubneh, Yibeltal Assefa, Kelemu Abebe.

**Formal analysis:** Belete Gelaw Walle, Bogale Chekole.

**Investigation:** Belete Gelaw Walle.

**Methodology:** Belete Gelaw Walle, Chalie Marew Tiruneh, Amare Kassaw.

**Software:** Belete Gelaw Walle, Amare Kassaw.

**Validation:** Belete Gelaw Walle, Nigusie Selomon, Yibeltal Assefa, Kelemu Abebe.

**Visualization:** Bogale Chekole, Yibeltal Assefa.

**Writing – original draft:** Belete Gelaw Walle.

**Writing – review & editing:** Nigusie Selomon, Chalie Marew Tiruneh, Moges Wubneh, Kelemu Abebe.

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
