## [Decision Letter · Decision Letter 0]

5 Sep 2023

PONE-D-23-13653Prevalence and determinants of under-nutrition among children on ART in Ethiopia: a systematic review and meta-analysisPLOS ONE

Dear Dr. Gelaw,

Thank you for submitting your manuscript to PLOS ONE. After careful consideration, we feel that it has merit but does not fully meet PLOS ONE’s publication criteria as it currently stands. Therefore, we invite you to submit a revised version of the manuscript that addresses the points raised during the review process.

We look forward to receiving your revised manuscript.

Kind regards,

Mulualem Endeshaw

Academic Editor

PLOS ONE

Journal Requirements:

- https://doi.org/10.1371/journal.pone.0261611

- https://doi.org/10.1186/s13690-021-00785-z

In your revision ensure you cite all your sources (including your own works), and quote or rephrase any duplicated text outside the methods section. Further consideration is dependent on these concerns being addressed.

3. Thank you for stating the following financial disclosure: "no"

Reviewers' comments:

Reviewer's Responses to Questions

**Comments to the Author**

1. Is the manuscript technically sound, and do the data support the conclusions?

Reviewer #1: Yes

Reviewer #2: Yes

2. Has the statistical analysis been performed appropriately and rigorously? 

Reviewer #1: Yes

Reviewer #2: Yes

3. Have the authors made all data underlying the findings in their manuscript fully available?

Reviewer #1: Yes

Reviewer #2: Yes

4. Is the manuscript presented in an intelligible fashion and written in standard English?

Reviewer #1: No

Reviewer #2: No

5. Review Comments to the Author

Reviewer #1: The submitted manuscript is a systematic review and meta-analysis. The topic of the review may be of scientific interest if adequately conducted, but for that, the review needs substantial improvement. In the present form, the review not only lacks any scientific added value but even cannot assure the scientific standard of the systematic review and meta analysis.

Specific comments.

1) The keywords are poorly formulated.

2) The manuscript lacks consistent format and proper English at several places.

3) In the Introduction, several statements lack essential information

Methodology

4) It is unclear how gray literature and government reports will be systematically searched.

5) The search string is poor and there is no evidence that is has been tested in at least one of the databases. A pilot search should have been conducted in advance.

6) It is not justified why only cross-sectional studies was included in the review, since it aims to synthetize information on a relationship that may be causal.

7) A clearly formulated PECO statement with inclusion and exclusion criteria is missing.

8) Screening is inadequately described (e.g. full-text screening is not even mentioned), and it is not clear whether it was conducted by two independent reviewers.

9) Information on the piloting of the data extraction sheet is missing.

10) Risk of bias assessment is missing. It is not clear how "quality" of the included publications was assessed.

10) The population, intervention, comparator and outcome (PICO) framework is a useful tool to consider were not clearly stated in the review.

11) The quality and bias assessment of included studies that are not described in detail in the review should be described in this section (for example, what type of quality assessment tools the authors intend to use....)

12) The eligibility criteria for selecting studies for inclusion in the systematic review were not clearly stated in the review; they should be written in detail.

13) Both primary and secondary outcomes should be clearly defined in the review

Result

Results presentation is not presented in the way readers can understand. Just a list of sentences without creativity. The authors should read previously published similar article and represent the results part clearly.

Discussion

The discussion part is not presented in the way guide the readers and clear way. In the first part of the discussion part, the authors should summarize the key findings which they are going to discuss. However, the authors just stated their discussion by comparing their finding with the previous findings. Even the way they are comparing with previous finding is not clear and just a list of results of previous study. Thus, the authors should read the previous study or any scientific writing books and try to summarize the key findings in the first paragraph of the discussion and then start results interpretation, and compare and contrast with previous studies. I strongly recommend for the authors should follow the previous study and try to present the discussion. In fact, not only the discussion whole manuscript should be written in the scientific principles (Clarity, brevity, chronological flow and attractiveness). For this following the previous quality article is very important

Reviewer #2: Thank you for your effort and commitment to summarize primary studies and generate the pooled result on important issues. I have the following concerns to be addressed before the endorsement for publication.

1. the manuscript lacks line number which impose difficulties to locate comments easily

2. the introduction should be revised in precise and to the point manner

3. The topic is addressed with the recent published paper with the same topic in sub-Saharan Africa https://www.ncbi.nlm.nih.gov/pmc/articles/PMC8728950/ where Ethiopia is located and studies conducted in Ethiopia also combined together to give the overall picture related to the topic. so, what is your motivation to conduct this review?

4. Abstract: HIV/AIDS infected children are at high risk to develop under-nutrition….better to start as ….Children infected with HIV are at ……

5. ‘The national burden of under-nutrition and associated factors among HIV/AIDS infected children is context, and needs evidence to renew future goal’…. this statement is not clear and needs revision and rephrasing

6. the introduction better to start the introduction with definitions of undernutrition

7. Stunting, under-weight, and wasting were highly prevalent…put the meaning of each terms in bracket like stunting (meaning), wasting (meaning)…

8. The second outcome to identify factors……the second objective or outcome?

9. Study design: All observational study designs reporting the prevalence of under-nutrition were eligible for this meta-analysis….what about the HIV/AIDS infection?

10. your target groups should be mentioned in exclusion criteria and study design part…among children on ART

11. Data selection process…. better to write ‘data extraction process’

12. For the second outcome variable???

13. In the sensitivity analysis, there is no study away from the lower and upper limit of confidence interval…here you have to mention at least the range of the pooled prevalence through exclusion of each study in the sensitivity analysis

6. PLOS authors have the option to publish the peer review history of their article (what does this mean?). If published, this will include your full peer review and any attached files.

Reviewer #1: **Yes: **SISAY ABEBE DEBELA

Reviewer #2: No

---

## [Author Response · Author response to Decision Letter 0]

27 Sep 2023

Comments to the Author

The authors of this systematic review and meta-analysis have presented useful data to determine the pooled national burden of under-nutrition among children on ART in Ethiopia. However, there are specific comments that the reviewer would like the authors to address for further improvement of our manuscript.

Authors’ response: we are very happy to the reviewer’s appreciation of our efforts and commitment; and we have just provided our respective responses to each of the specific reviewer comments and concerns detailed below. Additionally, we have addressed the additional journal requirements with in our document accordingly.

Reviewer #1 

1. The keywords are poorly formulated.

Thank you. We have amended accordingly. Please see line 68. 

2. The manuscript lacks consistent format and proper English at several places.

We have revised the manuscript to ensure that all formats are consistent and after repeated proof-reading of the manuscript, we have tried our best to edit the manuscript for English language usage. These changes are found throughout the revised version of our manuscript. 

3. In the Introduction, several statements lack essential information

We have carefully reviewed our manuscript based on your comment and we have revised some statements to be more informative. The changes can be appreciated from the tracked insertions and deletions on introduction section of the revised version manuscript.

Methodology

4. It is unclear how gray literature and government reports will be systematically searched.

Thank you. We have taken your constructive comment and amended our manuscript accordingly. Please see line 128 to 129. 

5. The search string is poor and there is no evidence that is has been tested in at least one of the databases. A pilot search should have been conducted in advance.

Sure! Nevertheless many studies have been done on this topic, we have done a comprehensive search in databases which include most peer-reviewed African journals.

6. It is not justified why only cross-sectional studies was included in the review, since it aims to synthetize information on a relationship that may be causal.

We would like to say sorry for the unclear expression we caused with respect to the articles included to this study, this review hasn’t been incorporated only “cross-sectional studies’’, but also other observational studies like “cohort studies” conducted among HIV-positive children in Ethiopia were included. Revisions have been made based on the comment. Please see line 159 to 160.

7. A clearly formulated PECO statement with inclusion and exclusion criteria is missing.

Thank you. We have made the necessary corrections accordingly. Please see line 128 to 134.

Screening is inadequately described (e.g. full-text screening is not even mentioned), and it is not clear whether it was conducted by two independent reviewers.

We have revised this part accordingly. Please see line 189 to 192.

8. Information on the piloting of the data extraction sheet is missing.

Thank you. We have taken the comment and made the necessary corrections accordingly. Please see from the data extraction part. 

9. Risk of bias assessment is missing. It is not clear how "quality" of the included publications was assessed.

Accepting the given comment, we have made the necessary corrections as tracked on page 11, of the revised version manuscript.

10. The population, intervention, comparator and outcome (PICO) framework is a useful tool to consider were not clearly stated in the review. 

Thank you very much for your comment. Appropriate revisions have been made accordingly. Please see line 165 to 169. 

11. The quality and bias assessment of included studies that are not described in detail in the review should be described in this section (for example, what type of quality assessment tools the authors intend to use....)

The methodological and evidence quality of the included studies were critically appraised using the modified Newcastle-Ottawa quality assessment tool scale adapted for observational studies. The quality of all the included studies was graded as ‘high quality’ because their numerical rating in Newcastle-Ottawa quality assessment tool scale was ≥6.

We have amended the manuscript accordingly. Please see this section.

12. The eligibility criteria for selecting studies for inclusion in the systematic review were not clearly stated in the review; they should be written in detail.

Thank you .We have revised this section accordingly.

13. Both primary and secondary outcomes should be clearly defined in the review

Thank you. We have made the necessary correction and all the improved changes can be observed from the tracked changes on methodology section of the revised version manuscript.

Result

14. Results presentation is not presented in the way readers can understand. Just a list of sentences without creativity. The authors should read previously published similar article and represent the results part clearly.

We have revised the result parts accordingly and amendment to the original manuscript can be found from the tracked insertions and deletions from the result section of the revised version manuscript.

Discussion

15. The discussion part is not presented in the way guide the readers and clear way. In the first part of the discussion part, the authors should summarize the key findings which they are going to discuss. However, the authors just stated their discussion by comparing their finding with the previous findings. Even the way they are comparing with previous finding is not clear and just a list of results of previous study. Thus, the authors should read the previous study or any scientific writing books and try to summarize the key findings in the first paragraph of the discussion and then start results interpretation, and compare and contrast with previous studies. I strongly recommend for the authors should follow the previous study and try to present the discussion. In fact, not only the discussion whole manuscript should be written in the scientific principles (Clarity, brevity, chronological flow and attractiveness). For this following the previous quality article is very important

Thank you so much. Improvement to the original manuscript can be appreciated from the tracked insertions and deletions of the discussion part in the revised version of our manuscript. 

Reviewer #2: 

1. The manuscript lacks line number which impose difficulties to locate comments easily

The line number has now been inserted to the manuscript. You can appreciate it throughout the revised version of our manuscript. 

2. the introduction should be revised in precise and to the point manner

We found the comment with paramount significance. Hence, the introduction parts of our manuscript has been heavily edited based on the given comments. Amendment to the original manuscript can be noticed from the tracked insertions and deletions on introduction section of the revised version of our manuscript. 

3. The topic is addressed with the recent published paper with the same topic in sub-Saharan Africa https://www.ncbi.nlm.nih.gov/pmc/articles/PMC8728950/ where Ethiopia is located and studies conducted in Ethiopia also combined together to give the overall picture related to the topic. so, what is your motivation to conduct this review?

Thank you very much. Although studies have been done on this topic, the inclusion and exclusion criteria of studies were completely different from our study. For instance, the studies published from 2010 to 2021 were included in studies done from SSA countries but we include studies published beyond this period to get further additional data. Furthermore, in Ethiopia, studies revealed the prevalence of undernutrition with great inconsistent across different geographical regions of the country. Besides the national prevalence, identifying modifiable risk factors of undernutrition among children living with HIV particularly in Ethiopia is a critical issue. Current and up-to-date information regarding the current burden of undernutrition in HIV-positive children using currently available nationwide studies is essential for clinicians, managers, and local policy makers to take appropriate actions. In this review, all observational studies reporting the prevalence of undernutrition among children living with HIV in Ethiopia were included to determine the current pooled national burden of undernutrition and its associated factors. 

4. Abstract: HIV/AIDS infected children are at high risk to develop under-nutrition….better to start as ….Children infected with HIV are at ……

We have revised the sentence accordingly. Please see line 25-26.

5. ‘The national burden of under-nutrition and associated factors among HIV/AIDS infected children is context, and needs evidence to renew future goal’…. this statement is not clear and needs revision and rephrasing

Thank you. The whole sentence has been revised and updated. Please see line 28-31.

6. the introduction better to start the introduction with definitions of undernutrition

We have made the necessary corrections accordingly. Please see line 73-75.

7. Stunting, under-weight, and wasting were highly prevalent…put the meaning of each terms in bracket like stunting (meaning), wasting (meaning)…

Thank you. We have made the necessary corrections as tracked on page 4, paragraph 1 of the revised version of our manuscript.

8. The second outcome to identify factors……the second objective or outcome?

We have made the necessary modifications as per the suggestion which is indicated from track change on page 8, paragraph 1 of the revised version of our manuscript.

9. Study design: All observational study designs reporting the prevalence of under-nutrition were eligible for this meta-analysis….what about the HIV/AIDS infection?

 We have amended the eligible section of our manuscript accordingly. Please see line 172 to 183.

10. your target groups should be mentioned in exclusion criteria and study design part…among children on ART

We have revised this part of the manuscript accordingly. Please see on page 9.

11. Data selection process…. better to write ‘data extraction process’

We have revised the sub-topic accordingly. accordingly. Please see line 195.

12. For the second outcome variable???

We have amended based on the suggestion. Please see on page 8. 

13. In the sensitivity analysis, there is no study away from the lower and upper limit of confidence interval…here you have to mention at least the range of the pooled prevalence through exclusion of each study in the sensitivity analysis

Thank you so much. We have revised the sentence, and the whole paragraph has also been updated accordingly. Please see line 286-292.

---

## [Decision Letter · Decision Letter 1]

2 Oct 2023

PONE-D-23-13653R1Prevalence and determinants of under-nutrition among children on ART in Ethiopia: a systematic review and meta-analysisPLOS ONE

Dear Dr. Gelaw, 

Thank you for submitting your manuscript to PLOS ONE. After careful consideration, we feel that it has merit but does not fully meet PLOS ONE’s publication criteria as it currently stands. Therefore, we invite you to submit a revised version of the manuscript that addresses the points raised during the review process.

We look forward to receiving your revised manuscript.

Kind regards,

Mulualem Endeshaw

Academic Editor

PLOS ONE

Journal Requirements:

Reviewers' comments:

Reviewer's Responses to Questions

**Comments to the Author**

1. If the authors have adequately addressed your comments raised in a previous round of review and you feel that this manuscript is now acceptable for publication, you may indicate that here to bypass the “Comments to the Author” section, enter your conflict of interest statement in the “Confidential to Editor” section, and submit your "Accept" recommendation.

Reviewer #1: All comments have been addressed

Reviewer #2: All comments have been addressed

2. Is the manuscript technically sound, and do the data support the conclusions?

Reviewer #1: Yes

Reviewer #2: Yes

3. Has the statistical analysis been performed appropriately and rigorously? 

Reviewer #1: Yes

Reviewer #2: Yes

4. Have the authors made all data underlying the findings in their manuscript fully available?

Reviewer #1: Yes

Reviewer #2: Yes

5. Is the manuscript presented in an intelligible fashion and written in standard English?

Reviewer #1: Yes

Reviewer #2: No

6. Review Comments to the Author

Reviewer #1: (No Response)

Reviewer #2: Dear authors

Thank you for your effort to address my questions and comments on the first draft of your manuscript. Here i kindly ask your effort to edit and revise your manuscript to fulfil the standard English.

7. PLOS authors have the option to publish the peer review history of their article (what does this mean?). If published, this will include your full peer review and any attached files.

Reviewer #1: **Yes: **Sisay Abebe Debela

Reviewer #2: No

---

## [Author Response · Author response to Decision Letter 1]

12 Oct 2023

Comments to the Author

The authors of this systematic review and meta-analysis have presented useful data to determine the pooled national burden of under-nutrition among children on ART in Ethiopia. Conversely, there is minor revision that the reviewer would like the authors to revise for further improvement of our manuscript.

Authors’ response: we are very glad to the reviewer’s appreciation of our efforts and commitment; and we have just provided our respective responses to each of the specific reviewer comments and concerns detailed below. Additionally, we have addressed the additional journal requirements with in our document accordingly.

Journal Requirements:

Authors’ response: We have revised the reference section to ensure that whither it is complete and correct or not. As a result, we found that references are correct, complete, and consistent with the strength of evidence and we didn’t made any new citation as well as deletion or remove from list of references. 

Reviewers' comments:

Reviewer's Responses to Questions

Comments to the Author

1. If the authors have adequately addressed your comments raised in a previous round of review and you feel that this manuscript is now acceptable for publication, you may indicate that here to bypass the “Comments to the Author” section, enter your conflict of interest statement in the “Confidential to Editor” section, and submit your "Accept" recommendation.

Reviewer #1: All comments have been addressed

Reviewer #2: All comments have been addressed

2. Is the manuscript technically sound, and do the data support the conclusions?

Reviewer #1: Yes

Reviewer #2: Yes

3. Has the statistical analysis been performed appropriately and rigorously?

Reviewer #1: Yes

Reviewer #2: Yes

4. Have the authors made all data underlying the findings in their manuscript fully available?

Reviewer #1: Yes

Reviewer #2: Yes

5. Is the manuscript presented in an intelligible fashion and written in standard English?

Reviewer #1: Yes

Reviewer #2: No

6. Review Comments to the Author

Reviewer #1: (No Response)

Reviewer #2: Dear authors

Thank you for your effort to address my questions and comments on the first draft of your manuscript. Here i kindly ask your effort to edit and revise your manuscript to fulfil the standard English.

Authors’ response: Thank you very much. We have carefully reviewed the manuscript and corrected any minor grammatical, typographic errors, and /inconsistencies. The changes can be appreciated from the tracked insertions and deletions on the second revised version of our manuscript.

---

## [Editor Report · Decision Letter 2]

23 Apr 2024

Prevalence and determinants of under-nutrition among children on ART in Ethiopia: a systematic review and meta-analysis

PONE-D-23-13653R2

Dear Dr. Gelaw,

We’re pleased to inform you that your manuscript has been judged scientifically suitable for publication and will be formally accepted for publication once it meets all outstanding technical requirements.

Kind regards,

Miquel Vall-llosera Camps

Staff Editor

PLOS ONE

---

## [Editor Report · Acceptance letter]

30 Apr 2024

PONE-D-23-13653R2 

PLOS ONE

Dear Dr. Gelaw Walle, 

I'm pleased to inform you that your manuscript has been deemed suitable for publication in PLOS ONE. Congratulations! Your manuscript is now being handed over to our production team.

Kind regards, 

on behalf of

Dr. Miquel Vall-llosera Camps 

Staff Editor

PLOS ONE